# More online interaction, more stock liquidity:——Evidence from Chinese stock exchange online interaction platform

**Kun Zhang**[1,2,3], **Zhenyi Hu**[1], **Jianfei Shen**[4], **Yuanyuan Wang**[5]*

1 School of Economics, Hebei University of Economics and Business, Shijiazhuang, Hebei, China, **2** The Hebei Collaborative Innovation Center for Urban-rural Integrated Development, Shijiazhuang, Hebei, China, **3** Center for Urban Sustainability and Innovation Development, Hebei University of Economics and Business, Shijiazhuang, China, **4** Finance Department, Pt Shenhua Guohua Pembangkitan Jawa Bali, Serang, Banten Province, Indonesia, **5** School of Accounting, Hebei University of Economics and Business, Shijiazhuang, Hebei, China

* Wangyuanyuan201610@163.com

**Data Availability Statement:** All data are available from the he China CSMAR database. The data is a third party data. Everyone can access these datasets from https://data.csmar.com/.The authors

## Abstract

This paper investigates the impact of online interaction between investors and enterprises on stock liquidity, using data from A-share listed companies in China from 2010 to 2021. Firstly, our findings reveal that more frequent interaction leads to better stock liquidity, and this result remains consistent across various robustness tests. Secondly, we observe that the expected tenure of senior executives and the ratio of institutional investor ownership exert a significant moderating effect on this relationship. Thirdly, this effect varies across enterprises at different development stages and with different ownership structures, being more pronounced in growing and privately-owned companies. Furthermore, this paper finds an inverted U-shaped relationship between reply length and stock liquidity, indicating that excessively long replies may introduce noise and negatively affect liquidity. This study provides new insights into how online interactions can improve market efficiency and offers practical implications for corporate governance and investor relations.

## 1. Introduction

For a long time, investors have relied on information intermediaries such as official company websites, analysts, and stock forums to obtain information about companies. With the rapid development of Internet technology, social media has emerged and flourished. Social media affects the capital market information environment, information dissemination speed, investor information acquisition, and communication channels [1–4]. Social media has gradually become an important medium for investors to obtain information.

In order to improve the information efficiency of the capital market and enhance the information environment for individual investors, the Shenzhen Stock Exchange and Shanghai Stock Exchange in China launched investor interaction platforms in 2010 and 2013, respectively—Interaction Easy" and "Shanghai e-Interaction". To our knowledge, similar interactive

did not have any special access privileges that others would not have.

**Funding:** Funded by Science Research Project of Hebei Education Department BJS2023018 for this research.

**Competing interests:** NO authors have competing interests.

platforms do not exist in other countries. Unlike Chinese social media such as Stock Bar and Weibo, and American social media such as Twitter and Facebook, "Interaction Easy" and "Shanghai e-Interaction" are officially operated by Chinese stock exchanges. Investors can ask companies questions through the platform, and companies will answer these questions. The interactive content and process have legal effect and credibility, and are subject to official supervision. Investors can not only make inquiries to enterprises at any time but also participate in interviews organized by enterprises. Additionally, investors can browse the text information on the interactive platform at any time and optimize their own investment decisions by observing the behavior of others [5]. Unlike traditional information intermediaries, online interaction is a form of information release and interpretation dominated by individual investors and involves direct interaction between investors and enterprises. As for the impact of social media on the capital market, most of the current research mainly focuses on how online interaction (among investors) influence the capital market. Online interaction among investors is specifically manifested in investors spontaneously creating, sharing and commenting on some topics on the platform, which may affect the investment behavior of others and further affect the pricing efficiency of the capital market. There are few studies investigate impact of online interaction between enterprises and investors on the capital market. Therefore, this paper chooses "Interaction Easy" and "Shanghai e-Interaction" as the research objects, and empirically examines the impact of online interaction between enterprises and investors on the enterprise stock liquidity, which enriches the existing literature.

While "Interaction Easy" and "Shanghai e-Interaction" are undoubtedly exciting new sources of information for capital markets, it is unclear whether online platform interaction will alleviate the degree of information asymmetry between companies and investors and affect stock liquidity. On one hand, the online interactive platforms can broaden the channels for individual investors to obtain information, help investors understand information, and alleviate the degree of information asymmetry between investors and companies [6]. Enterprises use the online interaction platform to answer investors' questions, clarify false rumors, and interact with investors, which helps improve investors' ability to interpret information [7]. Additionally, investors can use the online interaction platform to ask questions to companies, fully utilizing the active supervision ability of individual investors, improving the transparency of corporate information, and thus enhancing the information efficiency of the capital market [8]. On the other hand, social media sometimes produces a certain amount of noise, which can confuse, distort, and even change the real information, thus affecting the behavior of investors. Scholars observed that due to investors' personal and environmental factors, social media sometimes generates social noise, which affects the behavior of investors and further impacts the capital market [9]. "Polite words" and "answers that are not asked" without substantive information often occur. A large amount of irrelevant information increases the difficulty of information processing for individual investors and reduces their information acquisition efficiency.

The existing literature has not reached a consensus on the impact of online interaction on capital market efficiency. In addition, most studies mainly focus on the interaction between investors and lack consideration of the interaction between investors and firms. In order to fill these research gaps, this paper collected data from A-share listed companies in the Chinese market from 2010 to 2021 to explore the impact of online platform interaction (between investor and company) on stock liquidity.

The main findings of this work are as follows: 1) We find that more frequent interaction leads to better stock liquidity. To address the potential endogeneity problem in this paper, we also conduct several robustness tests. The result remains unchanged after controlling for individual firm fixed effects, lagging the explanatory variables by one period, using the DID model,

and replacing the explained and explanatory variables. 2) There is a moderating effect of executives' expected tenure and investor structure on the relationship between online interaction and stock liquidity. The expected tenure of a company's executives directly affects the internal governance environment and organizational strategy of the company. The longer the tenure, the more responsible the executives are. At this time, the company's internal information is more transparent, and the explanations to investors are clearer and more timely, leading to better stock liquidity. Since institutional investors have better information capabilities, the online interactive platform mainly serves individual investors. Thus, we find that online interaction has a stronger effect among stocks with a low proportion of institutional investors. 3) We explored the heterogeneous impacts of online interaction on stock liquidity. We have observed that such an effect is stronger in enterprises in the growth stage and in private enterprises. The underlying mechanism may be that enterprises in the growth stage and in private lack a stable profit model, and the principal-agent problem is more prominent. At this time, the interaction between investors and enterprises can supervise and govern the company, improve the quality of information disclosure, reduce the space and motivation for executives to hide information, and ultimately affect the enterprise's stock liquidity. 4) We also explored the potential nonlinear relationship between online platform interaction and stock liquidity. We find that reply length has an inverted U-shaped effect on stock liquidity, which rises first and then decreases. When the reply length is too long, it produces a certain amount of "noise" and has a negative effect on stock liquidity.

This paper contributes to current literature in several ways. Firstly, while prior research has primarily focused on the impact of investors' social media interactions on the information efficiency of capital markets [10–13], it has often overlooked the influence of direct information interactions between investors and enterprises. This study examines the impact of online interactions between investors and enterprises on corporate stock liquidity, thus enriching the current body of literature. Secondly, the paper explores the moderating effects of executives' expected tenure and investor structure on the relationship between online interactions and stock liquidity. This analysis provides novel insights into how online interactive platforms can be utilized to enhance corporate governance and improve the management quality of listed companies [14]. Finally, previous studies have not reached a consensus on the impact of online interaction on capital market efficiency [6, 9]. This paper not only analyzes the linear correlation between online platform interaction and stock liquidity but also verifies the nonlinear (inverted U-shaped) relationship between these variables. The findings can explain the inconsistencies in previous research to some extent and offer practical implications. Overall our results help enterprises to effectively use online interactive platforms to appropriately respond to investor inquiries, improve information quality and management level, reduce information asymmetry between investors and enterprises, and ultimately improve the pricing efficiency of the capital market.

The remainder of this paper proceeds as follows. In Section 2: Theoretical Analysis and Research Hypotheses. Section 3: Sample Selection, Model Development, and Variable Definition. Section 4: Main Results and Robustness Checks. Section 5: Additional Tests. Section 6: Conclusion, summarizes the full text.

## 2. Theoretical analysis and research hypothesis

### 2.1. Information technology and stock market efficiency

With the rapid development of information technology, scholars have increasingly focused on its impact. They found that information technologies such as blockchain, artificial intelligence, and big data could promote information dissemination and reduce information asymmetry,

thus enhancing the efficiency of the stock market [15, 16]. Farboodi observed that with the rise of information technology, market participants had more channels to obtain information [10]. Some scholars found that the rapid development of science and technology had given rise to a series of financial social platforms, which provided places for investors to discuss, analyze, and disseminate views [11] The advent of information technology could not only improve the timeliness of data analysis and sharing but also enhance the information transparency of financial institutions, promote better risk management, and improve the pricing efficiency of capital markets. Yu et al.believed that information technology made it possible for market participants to communicate and interact with each other [17]. Effective interaction could promote information disclosure, which in turn leads to more accurate and effective pricing.

However, some scholars believe that the emergence of information technology exacerbates the risk of information contagion, and a small information disturbance may quickly penetrate into other markets, leading to abnormal stock price fluctuations [18]. Yu et al.found that social media derived from information technology could significantly amplify market sentiment, thus affecting stock prices and investor behavior [19]. Goldstein et al. found that due to the development of information technology, market participants may have more information, but this may also reduce the efficiency of their investment decisions [20]. Dessaint et al. observed that new technologies may provide very limited useful information, which may interfere with managers' access to correct information [21].

## 2.2. Social media and corporate stock liquidity

The traders with inferior information are unwilling to trade, resulting in poor stock liquidity [22]. Based on this, some scholars have begun to study the role of social media in the capital market. Social media disclosure can alleviate information asymmetry between investors and companies reduce transaction costs, affect corporate stock liquidity, reduce the risk of stock price crashes, improve the accuracy of analyst forecasts, and subsequently affect the pricing efficiency of the capital market [23–27]. Blankespoor showed that companies can widely expand their information dissemination channels through social media [3]. He found that using Twitter to release press statements to investors can reduce information asymmetry. Da et al. found that when a company's Google search volume increased, the company's stock price increased in the near term [28].

Social media provides a platform for online interaction, including communication, information sharing, and networking. Online interaction will generate "information," alleviate information asymmetry between enterprises and investors, and affect enterprise stock liquidity. Firstly, compared with Chinese unofficial social media such as "stock bars" and "Dong Fang Fortune", "Interaction Easy" and "Shanghai e-Interaction" are officially operated by the Chinese stock exchange, which have legal effect and credibility. They are characterized by wide coverage, strong timeliness, high information quality, and strong interactivity [7]. The online interactive platform helps individual investors broaden their access to information. Through these platforms, enterprises answer investors' questions, clarify false rumors, and help investors correctly interpret professional information. Thus, online interaction between enterprises and investors on the interactive platform can alleviate the degree of information asymmetry, improve investors' information processing abilities, and have a positive effect on enterprise stock liquidity. Secondly, individual investors can use the investor interaction platform to ask questions about companies and play a role in supervision and governance. Feng et al. analyzed financial media reports and company Weibo data of Chinese listed companies from 2011 to 2016, finding that social media can effectively play a supervisory role and improve corporate information transparency [8]. Gao et al. found that social media effectively pressures

executives, improves the quality of information disclosure, increases their performance sensitivity, reduces the risk of adverse selection, and thus improves enterprise stock liquidity [29].

However, sometimes social media may generate "noise". Firstly, a great deal of information generated in the interaction may weaken the quality of information [30]. The information on the platform is subjective, and "no information content" or "answers not asked" often occur. The management's answers may also not meet investors expectations for the disclosure of specific information, failing to alleviate information asymmetry, or they may strategically hide bad news for their own self-interest. At this time, the interaction generates "noise". Secondly, Zimmerman et al. observed that social media generates "social noise" (information received through social media influenced by the external environment), which affects investor behavior and can lead to wrong decisions [9]. Additionally, for individual investors, a large amount of irrelevant information increases the difficulty of information processing. Based on this, some scholars find that interaction on online platforms generates "noise", worsening the information environment of enterprises and aggravating the information asymmetry between enterprises and investors.

Whether the interaction on online platforms can affect stock liquidity essentially explores whether the interaction generates "information" or "noise". Can the interaction alleviate information asymmetry between investors and companies, thereby improving corporate stock liquidity? Although actual results show that both effects of online interaction may coexist, with the gradual development of Chinese security markets, the improvement of investors' abilities and awareness of rights protection, and the increase in punishment intensity, behaviors such as manipulating information disclosure on online interactive platforms are effectively controlled. To sum up, this paper conjectures that interaction can alleviate the degree of information asymmetry between enterprises and investors, affect investors' behavior, and finally improve company liquidity. Based on the above analysis, this paper puts forward the following hypothesis:

**H1: The online interaction has a positive effect on stock liquidity.**

## 2.3. Moderating effect analysis

The above analysis doesn't consider the heterogeneity of the effect of enterprise managers on online interaction. The executives' expected tenure directly affects the internal governance environment and organizational strategy of the company. If the management expects a longer term at the enterprise, they will be more responsible to investors and more committed to the future of the company. At this time, they will respond more actively, quickly, and objectively on the interactive platform, improving the information transparency of the company, and finally having a positive effect on the liquidity of the company. Existing literature mainly studies from the perspective of equity incentives. For example, executive shareholding can alleviate information asymmetry, improve the quality of information disclosure [31], and stimulate corporate innovation capabilities [14]. This paper intends to analyze the moderating effect of executives' expected tenure on the relationship between online interaction and stock liquidity. Based on the above analysis, this paper proposes the following hypothesis:2:

**H2: The longer the expected tenure of executives, the significant the effect of the online interaction on stock liquidity.**

In addition to the effect of executives, the shareholding ratio of institutional investors also has a moderating effect on the relationship between information interaction and stock liquidity. Compared with individual investors, institutional investors have an advantage in

information acquisition, which is an important indicator for examining the external information environment of enterprises. Institutional investors not only have the ability to obtain and process information [3], but they also participate in corporate governance through large shareholdings, playing a supervisory role and alleviating the problem of information asymmetry between investors and the company [32]. The online platform primarily serves individual investors, who typically have poorer information acquisition and interpretation skills. Individual investors have a greater demand for online information interaction than institutional investors. Therefore, when the ratio of institutional ownership is low, and the holdings are mainly by individual investors, they will ask questions to enterprises through the online interactive platform. This online information interaction between enterprises and individual investors will alleviate information asymmetry and improve the liquidity of the company. At this time, the effect of online information interaction on improving stock liquidity is more obvious. Based on the above analysis and considering the differences in information capabilities between individual investors and institutional investors, this paper puts forward Hypothesis 3:

**H3: The higher the ratio of institutional ownership, the smaller the effect of the online interaction on stock liquidity.**

## 2.4. Heterogeneity analysis

Research on the impact of online interaction has not yet reached a consistent conclusion. An important reason may be the heterogeneous impacts of online interaction on stock liquidity across enterprises at different development stages and with different ownership structures. Enterprises at different development stages exhibit varying financial statuses, degrees of information asymmetry, development strategies, organizational structures, and financing abilities [33], leading to different reactions to online interaction. This paper posits that the development stage of enterprises leads to different effects of online interaction on stock liquidity.

Enterprises in the growth stage not only lack a stable profit model and have a large demand for cash, facing relatively large financing constraints [34], but also suffer from concentrated power in the hands of a few executives, making the principal-agent problem more prominent. The internal organizational structure of these enterprises is not yet perfect [35]. At this stage, interaction between investors and enterprises through online platforms plays a supervisory and governance role, improving executives' information disclosure behavior and reducing the space and motivation for executives to hide information, ultimately improving enterprise stock liquidity. In the mature stage, enterprises have a stable and profitable business model and sufficient cash flow [34]. At this time, the development of the enterprise has a bright future, shareholders and creditors are relatively stable, and executives are more willing to voluntarily disclose company information [35]. Therefore, the effect of online interaction on the stock liquidity of enterprises is relatively limited at this stage. For companies in a stage of decline, the profit margins drop sharply, and shareholders and executives urgently need to transform and upgrade to find new development goals [35]. At this stage, external supervision is strong, and the risk of hidden information exposure is relatively high. To sum up, this paper argues that for companies in mature and declining stages, executives have less space and motivation to hide information, and the online interaction between investors and enterprises plays a relatively limited role.

Based on the above analysis, taking into account the different financial conditions, information asymmetry, principal-agent conflicts, and financing capabilities of enterprises at different stages, the effect of online information interaction on improving stock liquidity is different. This paper proposes Hypothesis 4:

**H4: The impact of online interaction on stock liquidity is more significant for companies in the growth stage.**

State-owned enterprises and non-state-owned enterprises have differences in social responsibility, profitability, financing ability and other aspects [36–38]. State-owned listed companies are mainly run by the government, and their development strategies are more dependent on the government. The online interaction has very limited impact on state-owned listed companies. For non-state-owned enterprises, online interaction has a great impact and can effectively change the external information environment of enterprises. Based on the above analysis, this paper puts forward Hypothesis 5:

**H5: The impact of online interaction on stock liquidity is more significant for non-state-owned enterprises.**

## 3. Sample selection model development and variable definitions

### 3.1. Sample selection

To test these five hypotheses, we collected data from the CSMAR database (The China Stock Market and Accounting Research database) and CNRDS database (the China Research Data Service Platform), from 2010 to 2021. The sample firms are Shanghai and Shenzhen A-share listed firms. This paper processes the data as follows: (1) Excluding financial listed companies; (2) Excluding ST and delisting during this period (3) Eliminate missing samples of main variables; (4) winsorize (1%) in the data, and finally 27,251 observations remain.

### 3.2. Model development

Our first research question asks, can the online interaction has a positive effect on stock liquidity.To test our central Hypothesis 1, we estimate the following model:

$$\text{Liq}_{i,t} = \beta_0 + \beta_1 \text{Reply}_{i,t} + \beta_2 \text{Controls}_{i,t} + \sum \text{Ind} + \sum \text{Year} + \varepsilon_{i,t} \tag{1}$$

Among them, the explained variable in the regression is stock liquidity (Liq), the core explanatory variable is online interaction (Reply), and controls a series of control variables that may affect the stock liquidity of enterprises; ε is the random error term of the model. In practice, there exist several methods to estimate the stock liquidity. Among them, the indicator ILLIQ is a simple but effective one, which is suitable for measuring the stock liquidity of the Chinese stock market and focus on the corporate financial field.

Our second research question examines the moderating effect of executives' expected tenure on the relationship between the online interaction and stock liquidity. To test Hypothesis 2, this paper constructs the model as follows,

$$Liq_{i,t} = \beta_0 + \beta_1 Reply_{i,t} + \beta_2 EGT_{i,t} + \beta_3 Reply_{i,t} * EGT_{i,t} + \beta_4 Controls_{i,t} + \sum Ind + \sum Year + \varepsilon_{i,t} \tag{2}$$

Among them, EGT is the executive expected tenure. Indeed, the information environment is a multifaceted and multidimensional concept that is likely associated with the senior executives. The expected tenure of senior executives not only directly affects the future development of senior executives, but also affects the internal governance environment and organizational strategy of the company. If the executive expected tenure can produce a moderating effect on the relationship between the online interaction and stock liquidity, then $\beta_3$ should be significantly positive.

Our final research question examines the impact of the ratio of institutional ownership on the relation between the the online interaction and stock liquidity. To test Hypothesis 3, we use the following specification:

$$Liq_{i,t} = \beta_0 + \beta_1 Reply_{i,t} + \beta_2 INSH_{i,t} + \beta_3 Reply_{i,t}*INSH_{i,t} + \beta_4 Controls_{i,t} + \sum Ind + \sum Year + \varepsilon_{i,t} \quad (3)$$

Among them, INSH is the share-holding ratio of institutional investor.

### 3.3. Variable definitions

**3.3.1. Stock Liquidity (Liq).** The dependent variable, Liq, is the enterprise stock liquidity, measured by the ILLIQ. The indicator ILLIQ is suitable for measuring the stock liquidity of the Chinese stock market and focus on the corporate financial field, which can be calculated by

$$ILLIQ_{i,t} = \frac{1}{D_{i,t}} \Sigma_{d=1}^{D_{i,t}} \frac{R_{i,t,d}}{VOL_{i,t,d}} *10^8 \quad (4)$$

$R_{i,t,d}$ indicates the return rate of enterprise i considering cash dividend reinvestment $VOL_{i,t,d}$ on the dth trading day of year t, and indicates the transaction amount of enterprise i on the dth trading day of year t, the unit is RMB million, which $D_{i,t}$ is the transaction of enterprise i in year t number of days.

We adopt the method and take the opposite number, thus, we obtain our Liq [39]. The larger the value of Liq, the better the stock liquidity of the enterprise. It is calculated as follows:

$$Liq = -ILLIQ. \quad (5)$$

**3.3.2. The online interaction (Reply).** This paper defines "Reply" as the measure of online interaction, which is the core explanatory variable of this paper, we adopt the method [7]. Companies reply to investors' questions on interactive platforms can reflect the interaction between listed companies and investors. "Reply" is calculated by the following function: *Reply = ln*(1+*Number of Reply*), where Number of Reply is the number of the companies' answers on the Shenzhen Stock Exchange's "Interaction Easy" and Shanghai Stock Exchanges "Shanghai e-Interaction" platforms.

**3.3.3. Executive Expected Tenure (EGT).** This paper defines "EGT" as the executive expected tenure, we adopt the method [40]. "EGT" is calculated by the following function:

$$EGT_{i,t} = (GT_{industry,t} - GT_{i,t}) + (Age_{industry,t} - Age_{i,t}) \quad (6)$$

$GT_{i,t}$ represents the tenure of senior executives of enterprises. $Age_{i,t}$ indicates the age of the executives of company. $GT_{industry,t}$ represents the average tenure of senior executives of all enterprises in the industry. $Age_{industry,t}$ indicates the average age of executives of all enterprises in the industry.

**3.3.4. Enterprise life cycle (Cycle).** In this paper, we adopt the method based on the positive (+) and negative (-) of the net cash flow of operation, investment and financing, divide the enterprise life cycle into three stages, which are growth stage, mature stage and decline stage [41]. The measurement methods of each variable are shown in Table A1 in the S1 Appendix. The results show that there are 12,405 enterprises in the growth stage, 9,372 enterprises in the mature stage, and 5,413 enterprises in the decline stage.

**3.3.5. Control variable.** In this paper, the following variables are selected to be controlled: company listing time (Age), total executive compensation (Salary), executive shareholding

**Table 1. Definition and description of research variables.**

| Variable type | Variable name | Variable symbol | Variable definitions |
|---|---|---|---|
| Explained variable | Stock liquidity | Liq | See the previous section for detailed measurement methods of indicators. |
| Core explanatory variable | Online interaction | Reply | See the previous section for detailed measurement methods of indicators. |
| Grouping variable | Development stage | Cycle | See the previous section for detailed measurement methods of indicators. |
| | Ownership structure | Soe | Dummy variable, 1 for state ownership, 0 otherwise |
| Moderator | Executive Expected Tenure | EGT | See the previous section for detailed measurement methods of indicators. |
| | The ratio of institutional investors ownership | INSH | Number of shares held by institutional investors/ total number of shares |
| Control variable | Time to market | Age | Enterprise listed years |
| | Total Executive Compensation | Salary | Take the natural logarithm of total executive compensation |
| | Executive shareholding ratio | Share | Number of shares held by executives/total number of shares |
| | Book-to-market ratio | MB | Total Assets/Total Market Value |
| | Enterprise size | Size | Logarithm the company's total assets |
| | Industry | Ind | Referring to the 2012 edition of the China Securities Regulatory Commission's industry classification, there are 18 industries in total, and dummy variables are set. |

Note: Table 1 reports the definition and description of research variables, including the dependent variable, the dependent variable, as well as control variables.

ratio (Share), book-to-market value ratio (Mb), enterprise size (Size), industry fixed effects (Ind) and year fixed effects (Year). The measurement methods of each variable are shown in Table 1.

## 3.4. Descriptive statistics and results

Table 2 provides details of the variables' descriptive statistics. We can see that the average Liq is 4.217, the maximum value is 6.869, and the minimum 1.099. The average value of stock liquidity is -0.189, the standard deviation is 0.86, the minimum value is -7.203, and the maximum value is -0.002. The the average value of online interaction and stock liquidity is basically consistent with previous research results. This paper calculates the variance inflation factor (VIF) of the model in the regression analysis, and the results show that the inflation factor of this study is far less than 10, which proves that there is no serious multicollinearity problem in this paper.

**Table 2. Descriptive statistics of main research variables.**

| Variable | Number of samples | Average value | Standard deviation | Minimum value | Maximum value |
|---|---|---|---|---|---|
| Liq | 27,251 | -0.189 | 0.860 | -7.203 | -0.002 |
| Reply | 27,251 | 4.217 | 1.250 | 1.099 | 6.869 |
| Soe | 27,251 | 0.313 | 0.464 | 0 | 1 |
| EGT | 27,251 | 0.033 | 3.899 | -9.483 | 9.377 |
| INSH | 27,251 | 0.421 | 0.248 | 0.003 | 0.901 |
| Age | 27,251 | 9.629 | 7.772 | 0 | 27 |
| Salary | 27,197 | 15.01 | 0.791 | 13.09 | 17.14 |
| Share | 27,251 | 0.085 | 0.149 | 0 | 0.622 |
| Mb | 27,251 | 0.592 | 0.259 | 0 | 1.166 |
| Size | 27,251 | 22.09 | 1.288 | 19.71 | 26.09 |

Note: Table 2 reports the summary statistics of all variables used in the paper. In order to take into account the accuracy and intuitiveness of the data, it is reserved to 3 digits after the decimal point by default, and the data that cannot be satisfied is reserved to 1 significant figure after the decimal point, the same below.

**Table 3. Regression results of online interaction and stock liquidity.**

|  | Liq | Liq |
|---|---|---|
|  | (1) | (2) |
| Reply | 0.067*** | 0.054*** |
|  | (0.004) | (0.004) |
| Age |  | 0.015*** |
|  |  | (0.001) |
| Salary |  | -0.036*** |
|  |  | (0.008) |
| Share |  | -0.221*** |
|  |  | (0.037) |
| MB |  | -0.402*** |
|  |  | (0.025) |
| Size |  | 0.139*** |
|  |  | (0.006) |
| _cons | -0.044 | -2.381*** |
|  | (0.055) | (0.126) |
| YEAR | control | control |
| Industry | control | control |
| N | 27 251 | 27 251 |
| r$^2$ | 0.065 | 0.125 |
| f | 63.334 | 110.983 |

Note: Table 3 reports the OLS results of the relationship between online interaction and stock liquidity, as well as including control variables.

*, ** and *** indicate significance at 10%, 5% and 1%.

# 4. Results and analysis

## 4.1. Effects of online interaction on stock liquidity

An OLS regression was run to test the Hypothesis 1, Table 3 reports the regression results. Columns (1) of Table 3 is the regression results, the results show that the online interaction (Reply) has a significant positive effect on stock quality. Then we control for size, age, share, salary, mb, year and industry. Columns (2) of Table 3 shows the results of the regression that the regression coefficient is 0.054, which is significantly positive at the 1% level. These results lend support to H1: The interaction between companies and investors on the Internet platform has a positive effect on stock liquidity.

## 4.2. Robustness checks

The above results show that online interaction significantly improves the stock liquidity of enterprises. However, this result may suffer from potential endogeneity problems. For example, the issue of reverse causality may arise, where it is not that more frequent interaction on online platforms improves stock liquidity, but rather that better stock liquidity attracts more investors, leading to more frequent interaction on online platforms. Secondly, there is the problem of omitted variables. For example, companies with more frequent positive media coverage may not only have higher stock liquidity but also attract higher investor attention, resulting in more interactions with listed companies on online platforms. Therefore, to address the potential endogeneity problems in this paper and to ensure robustness, the following methods are adopted: (1) controlling for individual fixed effects, (2) using lagged variables by one period, (3) employing a DID model, (4) replacing the explained variables, and (5) replacing the explanatory variables.

**Table 4. Robustness and endogeneity tests.**

|  | Liq | Liq | Liq | Liq2 | Liq | Liq | Liq |
|---|---|---|---|---|---|---|---|
|  | (1) | (2) | (3) | (4) | (5) | (6) | (7) |
| Reply | 0.052***<br>(0.006) |  |  | 0.635***<br>(0.026) |  |  |  |
| L. Reply |  | 0.005***<br>(0.001) |  |  |  |  |  |
| c.Treat#c.Post |  |  |  | 0.062**<br>(2.03) |  |  |  |
| Treat |  |  |  | -0.050*<br>(-1.71) |  |  |  |
| Post |  |  |  | 0.019***<br>(2.83) |  |  |  |
| Ask |  |  |  |  | 0.049***<br>(0.004) |  |  |
| Arate |  |  |  |  |  | 0.069*<br>(0.028) |  |
| Meanword |  |  |  |  |  |  | -0.057<br>(0.206) |
| Age | 0.121**<br>(0.047) | -0.001***<br>(0.000) | -0.003<br>(-1.24) | -0.170***<br>(0.005) | 0.016***<br>(0.001) | 0.015***<br>(0.001) | 0.015***<br>(0.001) |
| Salary | -0.067***<br>(0.015) | 0.007***<br>(0.002) | 0.012*<br>(1.65) | 0.041<br>(0.046) | -0.036***<br>(0.008) | -0.031***<br>(0.008) | -0.030***<br>(0.008) |
| Share | -0.512***<br>(0.076) | -0.010<br>(0.008) | 0.025*<br>(1.87) | 3.106***<br>(0.215) | -0.219***<br>(0.037) | -0.195***<br>(0.037) | -0.194***<br>(0.037) |
| MB | -0.778***<br>(0.035) | -0.033***<br>(0.006) | -0.105***<br>(-4.53) | 2.634***<br>(0.146) | -0.408***<br>(0.025) | -0.435***<br>(0.025) | -0.432***<br>(0.025) |
| Size | 0.304***<br>(0.014) | 0.024***<br>(0.001) | 0.061***<br>(4.69) | -1.506***<br>(0.035) | 0.141***<br>(0.006) | 0.149***<br>(0.006) | 0.148***<br>(0.006) |
| _cons | -5.573***<br>(0.371) | -0.665***<br>(0.029) | -1.455***<br>(-4.18) | 39.229***<br>(0.733) | -2.403***<br>(0.126) | -2.601***<br>(0.130) | -2.503***<br>(0.151) |
| Year | control | control | control | control | control | control | control |
| Industry | control | control | control | control | control | control | control |
| N | 27197 | 20436 | 9,856 | 27187 | 27197 | 27197 | 27187 |
| r² | 0.114 | 0.060 | 0.016 | 0.296 | 0.125 | 0.120 | 0.120 |

Notes: This table presents the results of robustness checks and endogeneity tests conducted on the relationship between online interaction and stock liquidity.

Control for firm fixed effects. Column (1) of Table 4 shows that when controlling for the firm fixed effect, the explanatory variable is significantly positive at the 1% level, indicating that the interaction of online platforms has a positive effect on stock liquidity. The results are robust.

Lag one-period. The model is as follows:

$$Liq_{i,t} = \beta_0 + \beta_1 Reply_{i,t-1} + \beta_2 Controls_{i,t-1} + \sum Ind + \sum Year + \varepsilon_{i,t-1} \qquad (7)$$

Table 4 column (2) reports that after lagging one-period, the explanatory variable is significantly positive at the 1% level, indicating that the interaction of online platforms still has a positive effect on stock liquidity. It also provides evidence for the core research hypothesis of this paper from the side.

The DID model. The Shenzhen Stock Exchange's "Interaction Easy" was officially launched in 2010, while the Shanghai Stock Exchange's "Shanghai e-Interaction" was officially launched in 2013, which provides a good exogenous impact environment for the adoption of the DID

model. Comparing whether there is a significant difference in the stock liquidity, take 2007–2012 as the sample period, the DID model is as follows:

$$Liq_{i,t} = \beta_0 + \beta_1 Treat_{i,t} + \beta_2 Post_{i,t} + \beta_3 Treat_{i,t}*Post_{i,t} + \beta_4 Controls_{i,t} + \sum Ind + \varepsilon_{i,t} \qquad (8)$$

Among them, Treat represents the sample attribute, if the company is listed on the Shenzhen Stock Exchange, it takes the value 1, otherwise it takes the value 0. Post indicates the time attribute, the value is 0 for years before 2010, and the value is 1 for years after 2010. The coefficients that this paper is mainly concerned with $Treat_{i,t}*Post_{i,t}$. Column (3) of Table 4 shows that $\beta_3$ is significantly positive at the 5% level. The results show that comparing with the company on the Shanghai Stock Exchange's, the stock liquidity of enterprises in Shenzhen Stock Exchange has been significantly improved. This conclusion provides robust evidence for the positive effect of online interaction on stock liquidity.

Changing the way of measuring stock liquidity. In this paper, we defined "Liq2" as the measure of the stock liquidity. "Liq2" is calculated by the following function: the ratio of the stock trading volume to the total number of tradable shares in the annual effective trading days. Column (4) of Table 4 shows that the explanatory variable is significantly positive at the 1% level, which shows that the higher the investor's attention, the more times the network platform interacts, which can effectively alleviate the information asymmetry between companies and investors, the higher the turnover rate of corporate stocks, the better the liquidity of corporate stocks.

Change the measurements of online platform interactions. In this paper, we also define "Ask", "Arate", and "Meanword" as the measure of the online interaction. "Ask" is calculated by the following function: $Ask = ln(1+Number\ of\ Ask)(10)$, where Number of Ask is the number of the investors' questions on the Shenzhen Stock Exchange's "Interaction Easy" and Shanghai Stock Exchange's "Shanghai e-Interaction" platforms. "Arate" is calculated by the following function: Arate = NumberReply/NumberAsk(11). "Meanword" is calculated by the following function: Meanword = ln(1+replyword)/Size (12), where replyword is the length of the reply. The more detailed the reply from the boss company on the online interactive platform, the clearer the explanation, and the higher the quality of the online platform interaction.

Column (5) of Table 4 shows that the explanatory variable is significantly positive at the 1% level, which shows that the higher the investor's attention, the more the number of questions on the online interaction platform, the more frequent the interaction between companies and investors on the platform, and the better the liquidity of corporate stocks. Column (6) of Table 4 shows that it is significantly positive at the 10% level. The more positive the reply which can effectively alleviate the degree of information asymmetry between the company and investors, and the better the liquidity of the company's stock. Column (7) of Table 4 shows that the explanatory variables are negative but not significant. This result shows that in the process of online interaction, the relationship between the length of corporate reply content and stock liquidity may not be a simple linear relationship, which needs to be further tested in the future.

## 4.3. Moderating effect analysis

The moderated effect of the management's expected tenure is shown in column (2) of Table 5. The explanatory variable and the interaction term (Reply*EGT) are significantly positive at the 1% level. This indicates that when managers expect a longer tenure, they consider the long-term development of the company, interact more frequently with investors, explain problems more clearly and credibly, and actively transmit more information to the outside world. This effectively alleviates the degree of information asymmetry between enterprises and investors. Therefore, Hypothesis 2 of this paper is verified.

**Table 5. Moderating effects of management expected tenure and institutional investors' ownership ratio.**

| Variable | Liq (1) | Liq (2) | Liq (3) | Liq (4) | Liq (5) |
|---|---|---|---|---|---|
| Reply | 0.054*** (0.004) | 0.053*** (11.94) | 0.044*** (9.91) | 0.067*** (12.71) | 0.055*** (11.92) |
| EGT | | -0.010** (-2.17) | | | |
| INSH | | | -0.327*** (-12.83) | | |
| SOE | | | | 0.133*** (3.55) | |
| State holding | | | | | -0.038 (-1.06) |
| Reply*EGT | | 0.003*** (3.04) | | | |
| INSH*Reply | | | -0.060*** (-3.69) | | |
| SOE*Reply | | | | -0.045*** (-5.32) | |
| State holding*Reply | | | | | -0.123 (-0.86) |
| Age | 0.015*** (0.001) | 0.016*** (19.56) | 0.015*** (18.14) | 0.017*** (19.59) | 0.015*** (18.94) |
| Salary | -0.036*** (0.008) | -0.035*** (-4.39) | -0.028*** (-3.49) | -0.035*** (-4.43) | -0.036*** (-4.52) |
| Share | -0.221*** (0.037) | -0.223*** (-6.00) | -0.457*** (-11.05) | -0.245*** (-6.54) | -0.243*** (-6.43) |
| MB | -0.402*** (0.025) | -0.401*** (-15.92) | -0.446*** (-17.55) | -0.389*** (-15.40) | -0.391*** (-15.30) |
| Size | 0.139*** (0.006) | 0.140*** (22.96) | 0.162*** (25.52) | 0.140*** (22.88) | 0.141*** (22.76) |
| _cons | -2.381*** (0.126) | -2.417*** (-18.93) | -6.763*** (-5.96) | -2.472*** (-19.45) | -2.401*** (-18.75) |
| YEAR | control | control | control | control | control |
| Industry | control | control | control | control | control |
| N | 27197 | 27,197 | 27,197 | 27,197 | 26,715 |
| r² | 0.125 | 0.008 | 0.019 | 0.127 | 0.127 |

Notes: This table highlights the moderating effects of management expected tenure and the institutional investors' ownership ratio on stock liquidity.

The moderated effect of the institutional investor shareholding ratio is shown in column (3) of Table 5. The results show that the interaction term (Reply*INSH) is significantly negative at the 1% level. This result indicates that when the shareholding ratio of institutions is low and individual investors are relatively high, online interaction can alleviate the information asymmetry between investors and enterprises, and improve stock market liquidity. This finding supports Hypothesis 3.

To examine the moderating effects of different institutional environments on the link between online interaction and stock market liquidity, we constructed two sets of measures to assess different institutional environments [42]. The first set is SOE, a dummy variable that equals 1 if a firm's controlling shareholder is a government entity, and 0 otherwise. Our second set of measures, State holding, is based on the percentage of firm shares owned by the state. These measures were important to our analyses because they not only captured the effect of

state influence but also reflected the complementary effect of online interaction on national administrative supervision.

The moderated effect of different institutional environments is shown in columns (4) and (5) of Table 5. The results show that the interaction term (Reply*Soe) is significantly negative at the 1% level, indicating that the effect of online interaction on stock liquidity mainly affects non-state-owned enterprises. This result suggests that government regulation and influence have a negative impact on the influence of online interaction on stock liquidity.

## 4.4. Heterogeneity test

To verify Hypothesis 4 in this paper, enterprises are divided into three groups according to their life cycle stages, and regression is conducted for each group. The specific regression results are shown in Table 6. Columns (1)–(3) respectively represent the enterprises at the growth stage, mature stage, and decline stage. In columns (1) and (2), the regression results show that the explanatory variables are significantly positive at the 1% level. In column (3), the explanatory variables are positive but not significant. Further, the Fisher inter-group difference test is carried out on the enterprises at the growth and mature stages. The results show that the explanatory variable for enterprises at the growth stage is significantly better than for those at the mature stage. This paper finds that the impact of online interaction on stock liquidity is more significant for companies at the growth stage. For enterprises at the growth stage, more interaction between investors and the enterprise leads to better stock liquidity. Thus, the impact of online interaction on stock liquidity varies with the life cycle of the company, mainly affecting enterprises at the growth stage. Therefore, Hypothesis 4 of this paper is verified.

To verify Hypothesis 5 in this paper, the samples are divided into a group of state-owned enterprises and a group of non-state-owned enterprises according to the ownership structure. The specific regression results are shown in Table 6. Columns (4)–(5) respectively represent

**Table 6. The regression results of online interaction, stock liquidity, enterprise life cycle and ownership structure.**

|  | Growth stage | Mature stage | Decline stage | State-owned | Non-state-owned |
|---|---|---|---|---|---|
|  | (1) | (2) | (3) | (4) | (5) |
| Reply | 0.101 *** | 0.008 *** | 0.005 | 0.072 *** | 0.012** |
|  | (0.009) | (0.002) | (0.003) | (0.006) | (0.004) |
| Age | 0.026*** | 0.001*** | 0.001* | 0.021*** | 0.008*** |
|  | (0.002) | (0.000) | (0.001) | (0.001) | (0.001) |
| Salary | -0.087*** | 0.005 | 0.003 | -0.058*** | -0.004 |
|  | (0.016) | 0.003) | (0.005) | (0.011) | (0.007) |
| Share | -0.345*** | -0.009 | 0.058* | -0.147*** | -0.301 |
|  | (0.071) | (0.017) | (0.027) | (0.044) | (0.174) |
| Mb | -0.643*** | -0.059 *** | -0.105 *** | -0.479 *** | -0.124 ** |
|  | (0.053) | (0.011) | (0.017) | (0.036) | (0.021) |
| Size | 0.270*** | 0.026*** | 0.042*** | 0.198*** | 0.045*** |
|  | (0.012) | (0.003) | (0.004) | (0.009) | (0.005) |
| _cons | -4.304 *** | -0.669 *** | -0.932 *** | -8.994 *** | -1.692 *** |
|  | (0.258) | (0.057) | (0.090) | (1.640) | (0.343) |
| Year | control | control | control | control | control |
| Ind | control | control | control | control | control |
| N | 12405 | 9372 | 5413 | 18691 | 8506 |
| r $^2$ | 0.213 | 0.047 | 0.038 | 0.024 | 0.020 |
| f | 95.845 | 13.557 | 6.134 | 13.008 | 5.349 |

Notes: Table 6 reports the heterogeneous results of enterprise life cycle and ownership structure influence online interaction and stock liquidity.

the state-owned and non-state-owned enterprises. In columns (4) and (5), the regression results show that the explanatory variables are significantly positive at the 1% level. Further, the Fisher inter-group difference test is carried out on the enterprises. The results show that the impact on non-state-owned enterprises is significantly better than on state-owned enterprises. The regression results indicate that, compared with state-owned enterprises, non-state-owned companies that actively respond to investors questions effectively alleviate the information asymmetry between enterprises and investors and improve stock liquidity. The interaction between investors and enterprises on the online platform has a more significant effect on non-state-owned enterprises.

## 5. Additional tests

Previous studies found that there is a simple linear relationship between online interaction and stock liquidity. This paper not only analyzes the linear correlation between online platform interaction and stock liquidity, but also further verifies the nonlinear relationship between online platform interaction and stock liquidity. Some scholars found that during the interaction process, some companies often provide irrelevant answers or selectively disclose information. In such cases, the interaction generates a certain amount of "noise," increasing the difficulty of information interpretation for individual investors, affecting the efficiency of their information acquisition, and reducing stock liquidity. The varying interpretation abilities of individual investors can intensify information asymmetry [7]. Currently, online interaction does not always alleviate the degree of information asymmetry between investors and enterprises. Therefore, this paper posits that there may be an inverted U-shaped relationship between online platform interaction and stock liquidity. To examine this hypothesis, we introduce the square of the interaction variables (Reply, Arate, Ask, Meanword) in our model.

As shown in column (1) of Table 7, the coefficient of the square of "Reply" is significantly negative at the 1% level. Furthermore, we use the Utest command in STATA to test for a U-shaped relationship between the two variables. However, the regression result does not pass the test, indicating that there is no nonlinear relationship between "Reply" and stock liquidity, but rather a simple linear relationship. This supports the core research hypothesis of this paper from another perspective.

As shown in column (2) of Table 7, the regression coefficient of "Ask" is -0.003, indicating that there is no significant nonlinear relationship between these variables. As shown in column (3) of Table 7, the regression coefficient of "Arate" is 0.125, which is not significant.

As shown in column (4) of Table 7, the regression coefficient of "Meanword" is -7.336, which is significant at the 10% level. Further, we use the Utest command in STATA to check for a nonlinear relationship, and the regression results pass the test. These results show that during the interaction process on the online platform, there is an inverted U-shaped relationship between the length of the reply content and stock liquidity. As the number of words increases, the degree of information asymmetry initially decreases to a minimum and then gradually increases. When the length of the reply content exceeds the inflection point, certain "noise" is generated, which negatively affects stock liquidity.

The reasons for the above results are as follows: the more detailed the replies from companies to investors on the online interactive platform, the more "information" is transmitted. This online interaction can effectively alleviate asymmetry between the company and investors, improving the company's stock liquidity. However, as the length of the reply content increases, the amount of interactive information transmitted also increases, making the interaction gradually more complex. At this point, certain "noise" is generated during the online platform interaction. This "noise" not only hinders individual investors from extracting useful

**Table 7. Regression results of online interaction and stock liquidity.**

|  | Liq | Liq | Liq | Liq |
|---|---|---|---|---|
|  | (1) | (2) | (3) | (4) |
| Reply | 0.115 *** (0.021) |  |  |  |
| Reply $^2$ | -0.008 ** (0.003) |  |  |  |
| Ask |  | 0.075 *** (0.018) |  |  |
| Ask $^2$ |  | -0.003 (0.002) |  |  |
| Arate |  |  | -0.096 (0.158) |  |
| Arate $^2$ |  |  | 0.125 (0.117) |  |
| Meanword |  |  |  | 2.707 * (1.108) |
| Meanword $^2$ |  |  |  | -7.336 * (2.889) |
| Age | 0.015*** (0.001) | 0.016*** (0.001) | 0.015*** (0.001) | 0.015*** (0.001) |
| Salary | -0.036*** (0.008) | -0.036*** (0.008) | -0.031*** (0.008) | -0.031*** (0.008) |
| Share | -0.221*** (0.037) | -0.219*** (0.037) | -0.195*** (0.037) | -0.193*** (0.037) |
| Mb | -0.406*** (0.025) | -0.411*** (0.025) | -0.435*** (0.025) | -0.433*** (0.025) |
| Size | 0.140*** (0.006) | 0.142*** (0.006) | 0.149*** (0.006) | 0.146*** (0.006) |
| _cons | -2.498 *** (0.133) | -2.455 *** (0.131) | -2.558 *** (0.136) | -2.686 *** (0.167) |
| YEAR | control | control | control | control |
| Industry | control | control | control | control |
| N | 27197 | 27197 | 27197 | 27197 |
| r $^2$ | 0.125 | 0.125 | 0.121 | 0.120 |
| f | 108.161 | 107.833 | 103.375 | 103.351 |

Notes: Table 7 reports the additional test on the online interaction and stock liquidity.

information but also reflects differences in information application capabilities among individual investors. With limited attention, the degree of information asymmetry among individual investors can weaken the positive effect of online platforms on stock liquidity.

## 6. Conclusions

This paper examined whether online interaction between investors and enterprises can improve stock liquidity. The main finding is that a higher frequency of online interaction leads to higher stock liquidity. The results, tested by lagged one-period processing, fixed effects model, DID model, and other methods, remain robust. Secondly, executives' expected tenure and investor structure have a moderating effect on the relationship between online interaction and stock liquidity. Thirdly, there are heterogeneous impacts of online interaction on stock liquidity across enterprises at different development stages and with different ownership structures. The effect is most pronounced for enterprises in the growth stage and non-state-owned

enterprises. These results can be seen in Table A2 in S1 Appendix. Finally, this paper not only analyzes the linear relationship between online platform interaction and stock liquidity but also verifies the nonlinear relationship between the two.

## 6.1. Theoretical implications

The paper makes several contributions. Firstly, the existing literature mainly focuses on the impact of social interaction among investors, ignoring the impact of information interaction between investors and enterprises on the capital market. Therefore, this paper studies the impact of online interaction (between investors and enterprises) on corporate stock liquidity. This paper enriches the existing literature on how social media affects the capital market [10–13]. Secondly, this paper analyzes the moderating effect of executives' expected tenure and investor structure on the relationship between online interaction and stock liquidity. It provides new ideas for strengthening corporate governance and improving the management quality of listed companies by using network interactive platforms in practice. This aligns with the recent findings [14], who examined the implications of executive characteristics on firm decisions but left the realm of online interactions largely uncharted. Thirdly, this paper examines the heterogeneous impacts of online interaction on stock liquidity across enterprises at different development stages and with different ownership structures. It offers new ideas for enterprises to use social media to improve stock liquidity based on their specific characteristics. This approach resonates with the work of Lv, who emphasized the importance of firm-specific characteristics in corporate finance but did not specifically address the nuances introduced by social media interactions [43]. Fourthly, this paper not only analyzes the linear correlation between online platform interaction and stock liquidity but also further verifies the nonlinear relationship between the two. This conclusion provides a complementary explanation to the two opposing views in the existing literature—whether social media provides "information" or "noise" [6, 7, 9, 44]. It also has practical implications. The conclusion of this study helps enterprises to rationally use online interactive platforms in practice, improve the quality of information and management, alleviate the degree of information asymmetry between investors and enterprises, and thus improve the pricing efficiency of the capital market.

## 6.2. Practical implications

First, the findings of this study guide enterprises to recognize the important role of social media in their stock liquidity. Traditional one-way information transmission methods can no longer meet the needs of investors. Enterprises should pay attention to the dynamics of investor information on social media platforms, understand the issues that investors care about, and respond to their questions in a timely and proactive manner. Strengthening interactive communication with investors can improve the external information environment of enterprises and reduce information asymmetry between investors and enterprises.

Second, the heterogeneity test of this paper finds that the interaction between network platforms and stock liquidity has a certain degree of heterogeneity. Private enterprises and those in the growth stage should focus on setting up efficient interactive platforms to provide investors with more accurate and targeted information, helping to maintain the efficiency of corporate capital market information.

Third, this paper finds that sometimes "noise" is generated in the process of interaction. Some enterprises' replies are unnecessarily long, lack substantive information, and have poor response timeliness. Additionally, some executives use online interactive platforms to manipulate information disclosure for self-interest. It is necessary to standardize the management of interactive platforms. Regulatory authorities should actively guide enterprises to improve the

quality of information disclosure and simultaneously conduct strict supervision of underperforming enterprises.

## 6.3. Discussion

Our findings reveal the impact of online interaction between investors and enterprises on corporate stock liquidity. The main finding of this paper is that a higher frequency of online interaction leads to better stock liquidity. This effect varies across enterprises at different development stages and with different ownership structures. Additionally, there may be a nonlinear relationship between online platform interaction and stock liquidity. We find that online interaction (measured by the number of companies' answers to investors' questions) has a linear positive effect on stock liquidity. However, online interaction (measured by the length of replies) has a nonlinear effect on stock liquidity. Specifically, when the reply length is too long, it produces a certain "noise" and has a negative effect on stock liquidity. The contradictory conclusions of previous studies may be caused by the different measurement methods of online interaction. This conclusion not only provides a complementary explanation to the two opposing views that the existing literature provides "information" or "noise" for social media but also has practical implications.

## 6.4. Limitations and future research directions

As with any study, our research is not without limitations. First of all, this paper may have endogeneity problems. It is possible that better stock liquidity attracts more investors' attention and discussion, which leads to more frequent interaction on the online platform, rather than the more frequent interaction improving stock liquidity. Therefore, this paper adopts the DID model and substitutes the dependent variable and independent variable to alleviate these concerns. These methods and indicators help to overcome the impact of measurement errors to a certain extent.

Secondly, this study uses a relatively direct measurement index, namely, the number of times enterprises reply to investors. While these indicators provide a basic understanding, they may lack the depth and sophistication that more advanced analytical tools and methods can achieve. In the future, with the continuous development of text analysis technology and the deepening of research, we should pay attention to the depth and accuracy of analysis and mining, and explore which interactive topics and content are the most effective in conveying valuable information. This presents an interesting and substantial opportunity for further research.

Finally, the application of the research conclusions in this paper to other countries is not tested in this study. Previous studies have found strong links between major social media across countries. This raises the possibility that our findings, while originating in the Chinese context, may have broader applicability to online platforms in other countries. However, due to data availability and the length of the paper, we have not verified this within the scope of this study.

## Supporting information

**S1 Appendix.**
(DOCX)

## Author Contributions

**Data curation:** Zhenyi Hu.

**Funding acquisition:** Kun Zhang.

**Project administration:** Kun Zhang.

**Supervision:** Yuanyuan Wang.

**Validation:** Yuanyuan Wang.

**Writing – original draft:** Zhenyi Hu.

**Writing – review & editing:** Jianfei Shen.

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
