## [Decision Letter · Decision Letter 0]

9 Nov 2023

PONE-D-23-35566More Online Interaction, More Stock Liquidity: ——Evidence from Chinese stock exchange online interaction platformPLOS ONE

Dear Dr. hu,

Thank you for submitting your manuscript to PLOS ONE. After careful consideration, we feel that it has merit but does not fully meet PLOS ONE’s publication criteria as it currently stands. Therefore, we invite you to submit a revised version of the manuscript that addresses the points raised during the review process.

The paper examines the impact of information interaction between investors and firms on market efficiency from the perspective of stock liquidity, using reaction data of enterprises to investors on the interaction platform of Shenzhen and Shanghai Stock Exchanges. The study finds that the more frequent interaction between companies and investors, the better the stock liquidity. Moreover, the expected tenure of senior executives and the shareholding ratio of institutional investors have a significant moderating effect on the above effects. The impact of information interaction on stock liquidity mainly acts on growth stage enterprises and private enterprises. Additionally, the study finds an inverted U-shaped relationship between reply length and stock liquidity in the process of online information interaction.

Overall, I find the study to be interesting and has the potential to be published in PLOS One, after carefully take my following comments into considerations. Currently, the paper is not professionally structured and there are many sloppiness across the manuscript (different fonts of references and main text, tables straddling two pages, no description on tables, etc.). These types of sloppiness need to be fixed before resubmission. I suggest you following the paper titled “Initial coin offerings and entrepreneurial finance: the role of founders’ characteristics” published in the Journal of Alternative Investment and Xu, J. et al. (2023) (Inherited trust and informal finance. Journal of Business Finance & Accounting) on structure your paper.

Since you use a Chinese context, there are several recent papers on China’s institutional quality that need to be discussed. You can also design a few tests to examine the moderating effects of different institutional environment on the link between investor interaction and stock market liquidity, for example using formalism developed in An, J., Hou, W., & Zhang, Y. (2019). China’s rule of law in New Era: the rise of regulation and formalism. Journal of Chinese Economic and Business Studies, 17(3), 313-318, and bureaucrats checks developed in An, J., Armitage, S., Hou, W., & Liu, X. (2020). Do checks on bureaucrats improve firm value? Evidence from a natural experiment. Accounting & Finance, 60(5), 4821-4844.

The interaction you specifically looking at is facilitated by technology. I therefore suggest you motivate your paper incorporating a discussion of technology and finance. An, J., & Rau, R. (2021). Finance, technology and disruption. The European Journal of Finance, 27(4-5), 334-345, offers you some good background to start with. Similarly, An, J., Hou, W., & Lin, C. (2022). Epidemic disease and financial development. Journal of Financial Economics, 143(1), 332-358, also offers some hints on technology and finance.

Why the tables a showing up multiple times? Why do not add notes to the tables to make them self-contained. I suggest you follow the following studies to format your paper.

Guo, S., & An, J. (2022). Does terrorism make people pessimistic? Evidence from a natural experiment. Journal of Development Economics, 155, 102817.An, J. (2020). Is there an employee-based gender gap in informal financial markets? International evidence. Journal of Corporate Finance, 65, 101737. Please submit your revised manuscript by Dec 24 2023 11:59PM. If you will need more time than this to complete your revisions, please reply to this message or contact the journal office at plosone@plos.org. Please include the following items when submitting your revised manuscript:A rebuttal letter that responds to each point raised by the academic editor and reviewer(s). You should upload this letter as a separate file labeled 'Response to Reviewers'.A marked-up copy of your manuscript that highlights changes made to the original version. You should upload this as a separate file labeled 'Revised Manuscript with Track Changes'.An unmarked version of your revised paper without tracked changes. You should upload this as a separate file labeled 'Manuscript'.

We look forward to receiving your revised manuscript.

Kind regards,

Jiafu An

Academic Editor

PLOS ONE

Journal Requirements:

3. Thank you for stating the following financial disclosure: "National Natural Science Foundation of China Youth Project NO.71803037; Hebei Provincial Higher Education University Humanities and Social Science Research Project: Youth Top Talents Project BJS2023018."

4. Thank you for stating the following in your Competing Interests section: "NO authors have competing interests".

7. Please amend your authorship list in your manuscript file to include authors zhenyi hu, Kun Zhang, Jianfei Shen, and Yuanyuan Wang. 

Reviewers' comments:

Reviewer's Responses to Questions

**Comments to the Author**

1. Is the manuscript technically sound, and do the data support the conclusions?

Reviewer #1: Partly

2. Has the statistical analysis been performed appropriately and rigorously? 

Reviewer #1: N/A

3. Have the authors made all data underlying the findings in their manuscript fully available?

Reviewer #1: Yes

4. Is the manuscript presented in an intelligible fashion and written in standard English?

Reviewer #1: No

5. Review Comments to the Author

Reviewer #1: In your literature review, we suggest incorporating the following papers that are relevant to your study:

- Bao, Z., & Huang, D. (2020). Gender differences in reaction to enforcement mechanisms: A large-scale natural field experiment.

- Bao, Z., & Huang, D. (2021). Shadow banking in a crisis: Evidence from FinTech during COVID-19. Journal of Financial and Quantitative Analysis, 56(7), 2320–2355.

- Chen, M., Li, N., Zheng, L., Huang, D., & Wu, B. (2022). Dynamic correlation of market connectivity, risk spillover and abnormal volatility in stock price. Physica A: Statistical Mechanics and Its Applications, 587, 126506.

- Chen, M., Wang, Y., Wu, B., & Huang, D. (2021). Dynamic analyses of contagion risk and module evolution on the SSE a-shares market based on minimum information entropy. Entropy, 23(4), 434.

Yu, D., & Huang, D. (2023a). Cross-sectional uncertainty and expected stock returns. Journal of Empirical Finance, 72, 321–340.

Yu, D., & Huang, D. (2023b). Option-Implied Idiosyncratic Skewness and Expected Returns: Mind the Long Run. Available at SSRN 4323748.

Yu, D., Huang, D., & Chen, L. (2023). Stock return predictability and cyclical movements in valuation ratios. Journal of Empirical Finance, 72, 36–53.

These papers can provide valuable insights into the impact of information disclosure and interaction on market efficiency, as well as the role of social media in the capital market. For instance, Bao and Huang (2020) investigate the gender differences in reaction to enforcement mechanisms, which could be relevant to your study in understanding how different groups of investors may react to information interaction. Chen et al. (2022) explore the dynamic correlation of market connectivity, risk spillover, and abnormal volatility in stock price, which can help you better understand the relationship between information interaction and stock liquidity. By incorporating these papers into your literature review, you can provide a more comprehensive understanding of the existing research in this area and strengthen the theoretical foundation of your study.

In addition to improving the literature review, we have the following comments for you to address in your revision:

1. Please provide a more detailed explanation of the methodology used in your study, including the specific statistical tests and models employed. This will help readers better understand your approach and assess the validity of your findings.

2. Consider discussing the potential limitations of your study, such as the generalizability of your findings to other markets or the potential biases in the data collected from the interaction platforms.

3. It would be helpful to include a more in-depth discussion of the practical implications of your findings for both investors and firms. This can help readers better understand the real-world significance of your research.

Please address these comments in your revised manuscript, and we look forward to reviewing your updated submission.

References:

Bao, Z., & Huang, D. (2020). Gender differences in reaction to enforcement mechanisms: A large-scale natural field experiment.

Bao, Z., & Huang, D. (2021). Shadow banking in a crisis: Evidence from FinTech during COVID-19. Journal of Financial and Quantitative Analysis, 56(7), 2320–2355.

Chen, M., Li, N., Zheng, L., Huang, D., & Wu, B. (2022). Dynamic correlation of market connectivity, risk spillover and abnormal volatility in stock price. Physica A: Statistical Mechanics and Its Applications, 587, 126506.

Chen, M., Wang, Y., Wu, B., & Huang, D. (2021). Dynamic analyses of contagion risk and module evolution on the SSE a-shares market based on minimum information entropy. Entropy, 23(4), 434.

Yu, D., & Huang, D. (2023a). Cross-sectional uncertainty and expected stock returns. Journal of Empirical Finance, 72, 321–340.

Yu, D., & Huang, D. (2023b). Option-Implied Idiosyncratic Skewness and Expected Returns: Mind the Long Run. Available at SSRN 4323748.

Yu, D., Huang, D., & Chen, L. (2023). Stock return predictability and cyclical movements in valuation ratios. Journal of Empirical Finance, 72, 36–53.

6. PLOS authors have the option to publish the peer review history of their article (what does this mean?). If published, this will include your full peer review and any attached files.

Reviewer #1: No

---

## [Author Response · Author response to Decision Letter 0]

28 Dec 2023

Dear Editors and Reviewers:

Thank you for your letter and for the reviewers’ comments concerning our manuscript. Those comments are all valuable and very helpful for improving our paper, as well as the important guiding significance to our research. We have read comments carefully and have made revisions which we hope meets with approval. In what follows, we list each comment (in italics) and our response (in roman text). We also highlight the revised content in BLUE in the revised paper.

1.Since you use a Chinese context, there are several recent papers on China’s institutional quality that need to be discussed. You can also design a few tests to examine the moderating effects of different institutional environment on the link between investor interaction and stock market liquidity, for example using formalism developed in An, J., Hou, W., & Zhang, Y. (2019). China’s rule of law in New Era: the rise of regulation and formalism. Journal of Chinese Economic and Business Studies, 17(3), 313-318, and bureaucrats checks developed in An, J., Armitage, S., Hou, W., & Liu, X. (2020). Do checks on bureaucrats improve firm value? Evidence from a natural experiment. Accounting & Finance, 60(5), 4821-4844.

Response: This is a very meaningful suggestion. In order to examine the moderating effects of different institutional environment on the link between the online interaction and stock market liquidity, we referred to An, J., Armitage, S., Hou, W., & Liu, X. (2020), constructed two sets of measures on the extent to measure different institutional environment. The first set is SOE, which was a dummy variable that equals 1 if a firm’s controlling shareholder is a government entity, and 0 otherwise. Our second set of measures, State holding, was based on the percentages of firm shares owned by the state. The regression results show that the effect of online interaction on stock liquidity mainly affects non-state-owned enterprises. These results can be seen in our new Table 6. 

Position in the paper: See section 4.3 “Moderating effect analysis”.

2.The interaction you specifically looking at is facilitated by technology. I therefore suggest you motivate your paper incorporating a discussion of technology and finance. An, J., & Rau, R. (2021). Finance, technology and disruption. The European Journal of Finance, 27(4-5), 334-345, offers you some good background to start with. Similarly, An, J., Hou, W., & Lin, C. (2022). Epidemic disease and financial development. Journal of Financial Economics, 143(1), 332-358, also offers some hints on technology and finance.

Response: Thank you for this valuable comment. I agree that the interaction I'm focusing on is significantly influenced by technology. I have incorporated a discussion on technology and finance in my paper, and added them to our literature review part. 

'Position in the paper: See section 2.1 “Information technology and Stock market efficiency”.

3.Why the tables a showing up multiple times? Why do not add notes to the tables to make them self-contained. I suggest you follow the following studies to format your paper.

Response: Thanks for your suggestions! We have added the notes and format the tables. See from Table 1 to Table 8. 

4. In your literature review, we suggest incorporating the following papers that are relevant to your study:

- Bao, Z., & Huang, D. (2020). Gender differences in reaction to enforcement mechanisms: A large-scale natural field experiment.

- Bao, Z., & Huang, D. (2021). Shadow banking in a crisis: Evidence from FinTech during COVID-19. Journal of Financial and Quantitative Analysis, 56(7), 2320–2355.

- Chen, M., Li, N., Zheng, L., Huang, D., & Wu, B. (2022). Dynamic correlation of market connectivity, risk spillover and abnormal volatility in stock price. Physica A: Statistical Mechanics and Its Applications, 587, 126506.

- Chen, M., Wang, Y., Wu, B., & Huang, D. (2021). Dynamic analyses of contagion risk and module evolution on the SSE a-shares market based on minimum information entropy. Entropy, 23(4), 434.

Yu, D., & Huang, D. (2023a). Cross-sectional uncertainty and expected stock returns. Journal of Empirical Finance, 72, 321–340.

Yu, D., & Huang, D. (2023b). Option-Implied Idiosyncratic Skewness and Expected Returns: Mind the Long Run. Available at SSRN 4323748.

Yu, D., Huang, D., & Chen, L. (2023). Stock return predictability and cyclical movements in valuation ratios. Journal of Empirical Finance, 72, 36–53.

These papers can provide valuable insights into the impact of information disclosure and interaction on market efficiency, as well as the role of social media in the capital market. For instance, Bao and Huang (2020) investigate the gender differences in reaction to enforcement mechanisms, which could be relevant to your study in understanding how different groups of investors may react to information interaction. Chen et al. (2022) explored the dynamic correlation of market connectivity, risk spillover, and abnormal volatility in stock price, which can help you better understand the relationship between information interaction and stock liquidity. By incorporating these papers into your literature review, you can provide a more comprehensive understanding of the existing research in this area and strengthen the theoretical foundation of your study. Yu, D.,& Huang, D. (2023a) highlighted the need for effective information management and the importance of understanding the impact of social media on market sentiment and efficiency, explored how different levels of cross-sectional uncertainty can impact the expected returns on stocks, proposing that higher uncertainty might lead to higher expected returns as a compensation for the increased risk.

Response: Thanks for your suggestions! We have read the existing literature carefully. In terms of the relationship between the information technology and market efficiency, most existing literature shows that the information technology can improve the information disclosure, the stock market efficiency. Bao, Z., and Huang, D. (2021) explored that the rapid development of technology had given rise to fintech platforms, which provided an important venue for investors to discuss, analyze, and disseminate their views on market trends, risks, and opportunities. As a kind of social media, it could not only improve the timeliness of data analysis and sharing, but also improved the information transparency of financial institutions, promoted better risk management and improved the pricing efficiency of capital market. Chen, M., Li, N., Zheng, L., Huang, D., & Wu, B. (2022) explored how information shared across different market participants, facilitated by technological advancements and social media, contributed to market connectivity. This connectivity, while beneficial for the flow of information, also leaded to risk spillover, where the risk in one market segment could quickly affect others, contributing to abnormal stock price volatility. Chen, M., Wang, Y., Wu, B., & Huang, D. (2021) used the concept of minimum information entropy, highlighted the importance of efficient and transparent information disclosure in mitigating contagion risks particularly in the age of social media. Yu, D., & Huang, D. (2023b), found information was rapidly disseminated through various channels including social media, such market sentiments could be quickly amplified, impacting stock prices and investor behavior. Yu, D., Huang, D., & Chen, L. (2023) argued that better information disclosure and effective interaction among market participants leaded to more accurate and efficient pricing, as reflected in these valuation ratios.

Position in the paper: See section 2.1 “Information technology and Stock market efficiency”.

5.Please provide a more detailed explanation of the methodology used in your study, including the specific statistical tests and models employed. This will help readers better understand your approach and assess the validity of your findings.

Response: Thank you for this valuable comment. In response, we have enriched our manuscript with a more detailed explanation of the methodology. This includes a thorough elucidation of the variable selection, various statistical tests and analytical models we utilized. This enhanced methodological exposition aims to provide a clearer and more robust framework for understanding the scope and rigor of our analytical processes.

Position in the paper: See section 3 “Sample selection model development and variable definitions ”.

6.Consider discussing the potential limitations of your study, such as the generalizability of your findings to other markets or the potential biases in the data collected from the interaction platforms.

Response: Thank you for this valuable comment. Like any study, ours is not without limitations. Firstly, the application of this paper’s conclusions to other countries is not tested in the current study. The prior research detected an intense interconnection between the major social media cross countries. This raises the possibility that our findings, while derived from a Chinese context, may have broader applicability to online platforms in other countries. However, due to the limitation of data availability and article space, we have not verified these within the scope of our current research. Secondly, the methodological approach employed in this study utilized relatively straightforward measurement indicators. While these indicators provided a foundational understanding, they may lack the depth and sophistication achievable with more advanced analytical tools and methods. Lastly, in the future, we could develop a more comprehensive set of metrics, explored which type of interaction was most effective in conveying valuable information. This presented both a fascinating and substantial opportunity for further research.

Position in the paper: See section 9 “Limitations and future research directions”.

7.It would be helpful to include a more in-depth discussion of the practical implications of your findings for both investors and firms. This can help readers better understand the real-world significance of your research.

Response: This is a very meaningful suggestion. First, the findings of this study guide enterprises to pay attention to the important role of social media on its stock liquidity. Since, the traditional one-way information transmission method can no longer meet the needs of investors. Enterprises should pay attention to the dynamics of investor information on social media represented by interactive platforms, understand the issues that investors care about, and respond to investors' questions in a timely and proactive manner. Strengthen the interactive communication with investors, improve the external information environment of enterprises, and reduce the information asymmetry between investors and enterprises. 

Second, the heterogeneity test of this paper finds that the interaction between network platforms and stock liquidity has a certain degree of heterogeneity. Private enterprises and enterprises in growth stage should pay more attention to set up an efficient interactive platform to provide investors with more accurate and targeted information, which help to maintain the efficiency of corporate capital market information. 

Third, this paper finds that sometimes “noise” is generated in the process of interaction. Some enterprises’ reply is unnecessarily too long, without information, and poor response timeliness. Even more, some executives use the online interactive platform to manipulate information disclosure out of self-interest. It is necessary to standardize the management of interactive platforms. The supervision department should actively guide enterprises to improve the quality of information disclosure, and meantime conduct strict supervision on underperforming enterprises.

Position in the paper: See section 8 “Practical implications”.

We tried our best to improve the manuscript and made some minor changes in the manuscript. These changes will not influence the content and framework of the paper. We appreciate for Editors/Reviewers’ warm work earnestly, and hope that the correction will meet with approval. Once again, thank you very much for your comments and suggestions.

---

## [Decision Letter · Decision Letter 1]

12 Jan 2024

PONE-D-23-35566R1More Online Interaction, More Stock Liquidity: ——Evidence from Chinese stock exchange online interaction platformPLOS ONE

Dear Dr. hu,

Thank you for submitting your manuscript to PLOS ONE. After careful consideration, we feel that it has merit but does not fully meet PLOS ONE’s publication criteria as it currently stands. Therefore, we invite you to submit a revised version of the manuscript that addresses the points raised during the review process.

Here are summarized comments on the manuscript:

Reorganize the language in newly added Section 2.1 for improved logical flow.

Enhance the presentation of tables, ensuring that the sequence of rows of controls is clear.

Provide proper citations of existing literature in Section 7 (the contribution section).

Discuss endogeneity issues in Section 9 and propose potential solutions. Additionally, address potential biases in the data collected from interaction platforms, as raised in the previous report.

Consider grouping Sections 6-10 together to enhance overall readability.

Please also be reminded that comment regarding adding additional references are optional to follow.

We look forward to receiving your revised manuscript.

Kind regards,

Jiafu An

Academic Editor

PLOS ONE

Journal Requirements:

Reviewers' comments:

Reviewer's Responses to Questions

**Comments to the Author**

1. If the authors have adequately addressed your comments raised in a previous round of review and you feel that this manuscript is now acceptable for publication, you may indicate that here to bypass the “Comments to the Author” section, enter your conflict of interest statement in the “Confidential to Editor” section, and submit your "Accept" recommendation.

Reviewer #2: (No Response)

2. Is the manuscript technically sound, and do the data support the conclusions?

Reviewer #2: Yes

3. Has the statistical analysis been performed appropriately and rigorously? 

Reviewer #2: Yes

4. Have the authors made all data underlying the findings in their manuscript fully available?

Reviewer #2: Yes

5. Is the manuscript presented in an intelligible fashion and written in standard English?

Reviewer #2: Yes

6. Review Comments to the Author

Reviewer #2: Comments

1. The language of newly added Section 2.1 should be re-organized in a logic way.

2. The presentation of tables (such as table 5) should be improved. For instance, the sequence of rows of controls seems mixed.

3. Section7 (the contribution section) lacks proper citations of existing literature.

4. Please also discuss endogeneity issues in section 9 and suggest possible solutions, moreover, the potential biases in the data collected from the interaction platforms (as mentioned by last report) haven’t been discussed.

5. Sections 6-10 can be put together for better readability.

7. PLOS authors have the option to publish the peer review history of their article (what does this mean?). If published, this will include your full peer review and any attached files.

Reviewer #2: No

---

## [Author Response · Author response to Decision Letter 1]

30 Jan 2024

Dear Editors and Reviewers:

Thank you for your letter and for the reviewers’ comments concerning our manuscript. Those comments are all valuable and very helpful for improving our paper, as well as the important guiding significance to our research. We have read comments carefully and have made revisions which we hope meets with approval. In what follows, we list each comment (in italics) and our response (in roman text). We also highlight the revised content in BLUE in the revised paper.

1.The language of newly added Section 2.1 should be re-organized in a logic way.

Response: This is a very meaningful suggestion. We appreciated your feedback on Section 2.1. We have restructured it to ensure that the ideas are presented more logically and coherently. Thank you for pointing this out.

Position in the paper: See section 2.1“Information technology and Stock market efficiency”.

2.The presentation of tables （such as table 5） should be improved. For instance, the sequence of rows of controls seems mixed.

Response: Thanks for your suggestions! We have revisited the table and reorganized the sequence of the control rows to enhance clarity and coherence. 

Position in the paper: See from Table1 to Table 8.

3.Section7（the contribution section） lacks proper citations of existing literature.

Response: Thank you for this valuable comment. In response, we have thoroughly reviewed this section and incorporated appropriate citations to ensure our contributions are well-supported.

Position in the paper: See section 6.1“Theoretical implications”.

4. Please also discuss endogeneity issues in section 9 and suggest possible solutions, moreover, the potential biases in the data collected from the interaction platforms （as mentioned by last report） haven’t been discussed.

Response: Thank you for this valuable comment. In this current version we have discussed the endogeneity problems and limitations of this paper, and discuss the potential bias that We may have in data collection. We propose feasible solutions to alleviate these problems based on the methods of replacing explanatory variables and explain variables in the DID model. Your feedback helps to improve the rigor and comprehensiveness of our research.

Position in the paper: See section 6.4“Limitations and future research directions”.

5.Sections 6-10 can be put together for better readability.

Response: Thank you for the suggestion. We have restructured these sections and put them into one section. Consolidating these sections may indeed provide a more cohesive and comprehensive narrative.

Position in the paper: See section 6“Conclusions”.

We tried our best to improve the manuscript and made some minor changes in the manuscript. These changes will not influence the content and framework of the paper. We appreciate for Editors/Reviewers’ warm work earnestly, and hope that the correction will meet with approval.

Once again, thank you very much for your comments and suggestions.

---

## [Decision Letter · Decision Letter 2]

26 Jun 2024

PONE-D-23-35566R2More Online Interaction, More Stock Liquidity: ——Evidence from Chinese stock exchange online interaction platformPLOS ONE

Dear Dr. hu,

Thank you for submitting your manuscript to PLOS ONE. After careful consideration, we feel that it has merit but does not fully meet PLOS ONE’s publication criteria as it currently stands. Therefore, we invite you to submit a revised version of the manuscript that addresses the points raised during the review process.

**ACADEMIC EDITOR: Address the minor comments from reviewers.**==============================

We look forward to receiving your revised manuscript.

Kind regards,Muhammad Usman Tariq, Ph.DPFHEA, CFCIPD, CMBESFSEDA, SMIEEEAcademic EditorPLOS ONE 

Journal Requirements:

Reviewers' comments:

Reviewer's Responses to Questions

**Comments to the Author**

1. If the authors have adequately addressed your comments raised in a previous round of review and you feel that this manuscript is now acceptable for publication, you may indicate that here to bypass the “Comments to the Author” section, enter your conflict of interest statement in the “Confidential to Editor” section, and submit your "Accept" recommendation.

Reviewer #2: All comments have been addressed

Reviewer #3: All comments have been addressed

Reviewer #4: All comments have been addressed

Reviewer #5: All comments have been addressed

Reviewer #6: All comments have been addressed

Reviewer #7: All comments have been addressed

Reviewer #8: All comments have been addressed

Reviewer #9: All comments have been addressed

Reviewer #10: All comments have been addressed

2. Is the manuscript technically sound, and do the data support the conclusions?

Reviewer #2: Yes

Reviewer #3: Yes

Reviewer #4: Partly

Reviewer #5: Yes

Reviewer #6: Yes

Reviewer #7: Yes

Reviewer #8: Yes

Reviewer #9: Yes

Reviewer #10: Yes

3. Has the statistical analysis been performed appropriately and rigorously? 

Reviewer #2: Yes

Reviewer #3: Yes

Reviewer #4: Yes

Reviewer #5: N/A

Reviewer #6: Yes

Reviewer #7: I Don't Know

Reviewer #8: Yes

Reviewer #9: Yes

Reviewer #10: Yes

4. Have the authors made all data underlying the findings in their manuscript fully available?

Reviewer #2: Yes

Reviewer #3: No

Reviewer #4: Yes

Reviewer #5: Yes

Reviewer #6: Yes

Reviewer #7: Yes

Reviewer #8: Yes

Reviewer #9: Yes

Reviewer #10: Yes

5. Is the manuscript presented in an intelligible fashion and written in standard English?

Reviewer #2: Yes

Reviewer #3: Yes

Reviewer #4: (No Response)

Reviewer #5: Yes

Reviewer #6: Yes

Reviewer #7: Yes

Reviewer #8: Yes

Reviewer #9: No

Reviewer #10: Yes

6. Review Comments to the Author

Reviewer #2: Overall, this revision largely improves the quality of this paper. I suggest that autghor should check the typos/formatting problem in 6.1. Theoretical implications.

Reviewer #3: This paper examines an interesting and important topic and employs sound empirical analysis characterized by technical competence. The methodology is carefully designed and the analytical execution is both sound and insightful. Such a solid empirical foundation not only strengthens the findings of the study, but also increases its reliability and relevance to the field.

Moreover, the results have important policy implications, especially from the perspective of stock market regulation to enhance investor protection. The paper makes a noteworthy contribution to the existing literature and enriches our understanding of the online interaction between firms and investors and its consequences.

Reviewer #4: i just went through the paper, the authors has done most of the part what was suggested by the reviewers, i am OK and satisfied with the paper quality.

Reviewer #5: The paper seems like an interesting job. Thus, I consider the paper suitable for being published in the PLOS ONE. I support this research that is deemed a good contribution to the journal’s future achievements. The revision that has been made by the author is satisfactory. My decision is to accept the paper. Apparently, the current version of the manuscript has convinced me. I affirm the revised version. To me, no further revision is needed. I wish to express my sincere appreciation to the authors for their scientific efforts during the revision process. Great job. Congratulations!

Reviewer #6: Apparently the manuscript has been improved as per the given instructions of the previous reviewers. The work has been improved.

Reviewer #7: Thank you very much for your revision. I appreciate your responses to the previous comments and the substantial improvements made to the manuscript.

Reviewer #8: (No Response)

Reviewer #9: The authors adequately address the issues raised by the previous reviewer. However, these are a few issues when addressed by the authors will help the readers understand and appreciate the concept. These issues are stated below:

The abstract did not show what the problem authors want to solve, the approach they intend to delve into the problem, and how significant it is in investigating and solving the problem. The abstract gives adequate information to the reader about the research and its conclusion.

It is difficult to tell whether these are the research findings or study objectives being presented in the introduction. These make the paper complex for readers. If they represent gaps found in the literature, they should be properly cited, or if they are study objectives, they should be well stated.

The paper lacks clarity and has repetitive sentences. The paper needs a thorough reading to correct minor grammatical errors. Besides, research is about facts, but the authors say “We believe that as the managers of a company…” It sounds accurate if the authors can reframe the sentences.

The results of the hypothesis can be summarized by showing the accepted and rejected results in a table to enable readers to understand better.

Reviewer #10: No further modifications are required, all comments have been addressed. The manuscript can be published in the current form

7. PLOS authors have the option to publish the peer review history of their article (what does this mean?). If published, this will include your full peer review and any attached files.

Reviewer #2: No

Reviewer #3: No

Reviewer #4: No

Reviewer #5: **Yes: **Dorian Aliu

Reviewer #6: **Yes: **Dr. Muhammad Iftikhar Ali

Reviewer #7: No

Reviewer #8: **Yes: **Sayed Farrukh Ahmed

Reviewer #9: **Yes: **Akpamah Peter

Reviewer #10: No

---

## [Author Response · Author response to Decision Letter 2]

8 Jul 2024

Dear Editors and Reviewers:

Thank you for your letter and for the reviewers’ comments concerning our manuscript. Those comments are all valuable and very helpful for improving our paper, as well as the important guiding significance to our research. We have read comments carefully and have made revisions which we hope meets with approval. In what follows, we list each comment (in italics) and our response (in roman text). We also highlight the revised content in BLUE in the revised paper.

1.Overall, this revision largely improves the quality of this paper. I suggest that autghor should check the typos/formatting problem in 6.1. Theoretical implications.

Response: Thank you for recognizing the improvements in our paper. We have carefully reviewed and corrected the typos and formatting issues in section 6.1 Theoretical Implications. We appreciate your valuable feedback!

Position in the paper: See section 6.1 “Theoretical implications”.

2.The authors adequately address the issues raised by the previous reviewer. However, these are a few issues when addressed by the authors will help the readers understand and appreciate the concept. These issues are stated below:The abstract did not show what the problem authors want to solve, the approach they intend to delve into the problem, and how significant it is in investigating and solving the problem. The abstract gives adequate information to the reader about the research and its conclusion.It is difficult to tell whether these are the research findings or study objectives being presented in the introduction. These make the paper complex for readers. If they represent gaps found in the literature, they should be properly cited, or if they are study objectives, they should be well stated.The paper lacks clarity and has repetitive sentences. The paper needs a thorough reading to correct minor grammatical errors. Besides, research is about facts, but the authors say “We believe that as the managers of a company…” It sounds accurate if the authors can reframe the sentences.The results of the hypothesis can be summarized by showing the accepted and rejected results in a table to enable readers to understand better.

Response: Thanks for your valuable suggestions. Firstly, we revised the abstract to clearly state the problem we try to investigate and our research findings. Secondly, in the introduction, we rewrote research gaps, findings and contributions to enable readers to understand better. Finally, in order to make it easier for readers to read and understand this paper, we presented the main hypotheses and results of this paper in Appendix A2.

Position in the paper: See abstract, section 1 “Introduction” and Appendix A2. 

 We tried our best to improve the manuscript and made some minor changes in the manuscript. These changes will not influence the content and framework of the paper. We appreciate for Editors/Reviewers’ warm work earnestly, and hope that the correction will meet with approval.

Once again, thank you very much for your comments and suggestions.

---

## [Editor Report · Decision Letter 3]

18 Jul 2024

More Online Interaction, More Stock Liquidity: ——Evidence from Chinese stock exchange online interaction platform

PONE-D-23-35566R3

Dear Dr. hu,

We’re pleased to inform you that your manuscript has been judged scientifically suitable for publication and will be formally accepted for publication once it meets all outstanding technical requirements.

Kind regards,

Muhammad Usman Tariq, Ph.D

PFHEA, CFCIPD, CMBE

SFSEDA, SMIEEE

Academic Editor

PLOS ONE
---

## [Editor Report · Acceptance letter]

26 Jul 2024

PONE-D-23-35566R3 

PLOS ONE

Dear Dr. Hu, 

I'm pleased to inform you that your manuscript has been deemed suitable for publication in PLOS ONE. Congratulations! Your manuscript is now being handed over to our production team.

Kind regards, 

on behalf of

Dr. Muhammad Usman Tariq 

Academic Editor

PLOS ONE